# Bibliometric Analysis of Corporate Social Responsibility in Tourism

**Chanin Yoopetch** ⬤, **Suthep Nimsai** *⬤ **and Boonying Kongarchapatara**

Center for Research on Sustainable Leadership, College of Management, Mahidol University, Bangkok 73170, Thailand
* Correspondence: suthep.nim@mahidol.ac.th; Tel.: +66-22-062-000

**Abstract:** The large amount of research on corporate social responsibility in tourism shows its importance as a field of study. The role of tourism organizations and their impacts on sustainability have become increasingly important in recent decades. In addition, research on corporate social responsibility has expanded in scale and scope and can be found in a number of academic journals. The aim of this paper is to present the details of the academic work on corporate social responsibility in tourism and to demonstrate the intellectual structure of research in this field. This study analyzes 571 articles obtained from Scopus and published from 2002 to 2022 (August) and presents the development and growth of knowledge in corporate social responsibility and tourism. The study method used to extract the articles was based on the preferred reporting items for systematic reviews and meta-analyses (PRISMA). The results indicate that this field of study has expanded significantly from being studied primarily in Europe towards being studied also in developing countries, such as China and India. In addition, the research themes emerging in the field of corporate social responsibility in tourism include tourism behaviors and strategic approaches to corporate social responsibility. This review highlights the emerging trends in research on corporate social responsibility in tourism, the dominant academic journals, and the countries that focus on research in this area. Furthermore, directions for future research are also presented.

**Keywords:** corporate social responsibility; tourism; bibliometric analysis; sustainability





## 1. Introduction and Conceptual Background

Corporate social responsibility has become one of the most important topics in a variety of industries, including the tourism and hospitality industries, and is concerned with the values of all stakeholders, including social welfare and harmony, natural environments [1–3], and the social welfare and benefits of future generations. Furthermore, corporate social responsibility can help create consumer and employee trust [4,5], is one of the leading themes in current business environments, and is integrated as part of many corporations' strategic plans. Additionally, it is crucial that a company shows their commitments and responsibilities as a part of their sustainable development plans. However, based on most initiatives in corporate social responsibility from tourism organizations, the aims of tourism organizations are related to reducing cost savings and improving reputation rather than achieving the true outcome of corporate social responsibility [6].

Corporate social responsibility (CSR) includes all of the policies and practices that organizations implement to enhance the conditions and wellbeing of the participants, stakeholders, communities, and other parties in society in relation to the requirements, laws, traditions, and acceptance of the stakeholders [7].

Past studies on corporate social responsibility in tourism have branched out in various directions, with earlier work in this field tending to focus on macroperspectives [5,6,8] and later work exploring research more in terms of the microperspectives of corporate social responsibility in tourism. As the tourism industry has expanded, increasingly more

tourism businesses have emerged around the world, ranging from local to multinational companies. Furthermore, the tourism industry has a great impact on the usage of natural resources, human resources, and financial resources; therefore, ensuring that CSR processes and their implementation are effectively managed and monitored, including planning, impact assessment, resource allocation, monitoring, implementation, and evaluation [4–6], is important. In addition, CSR in tourism is represented by the responsibility of good corporate citizens of tourism organizations; as tourism develops, there are some potential negative consequences from the tourism activities and initiatives by the tourism organizations. For these reasons, the focus of CSR in tourism reflects the sustainable pathway of tourism developments and ethical behaviors to support the sustainable growth of the tourism industry [7,8].

Despite the ongoing COVID-19 pandemic around the world, several CSR initiatives were implemented to support society during this crisis. Several organizations [9], including tourism organizations, reflected on their social responsibility by in providing assistance to those affected by the pandemic, by employing or safeguarding workers despite the financial difficulties faced by the organization, and by creating many social programs to help stakeholders. CSR programs have evolved with changes in society; the economy; and the environment, including climate change. The social demands of stakeholders have particularly changed rapidly with recent new and emerging problems.

Therefore, CSR initiatives need to be updated and sensitive to the ever-changing environments in order to meet these new social needs and social goals, and firms should begin planning corporate social responsibilities with clear goals in mind and by using tangible measurements with respect to corporate social responsibility outcomes, such as fair income distribution, gender equality, and measurable improvements in environmental impacts from their corporate social responsibility [10]. CSR initiatives are expected to offer social and environmental benefits to the site of these CSR activities [11]. However, many studies in the past tended to focus on the CSR activities more than on the actual impacts of the CSR initiatives. The process behind these CSR initiatives includes its initial conception, activity development, implementation, and performance.

Generally, tourism development projects in private sectors are well supported by government agencies and public sectors for several reasons [12]: promoting employment, reducing income inequality, promoting the efficient use of resources, extending public transportation and city infrastructure, and developing worker skills. In addition, the development of the tourism industry can directly affect the development and improvement of traditions, cultures, festivals, and ways of life.

Tourism initiatives can result in both positive and negative impacts on the environment and society [12]. The role of CSR has been considered important not only for society and the environment but also for maximizing business returns [2]. In addition, CSR activities provide a positive image of the company by creating positive stakeholder relationships and thus promoting advocacy in their stakeholders.

CSR activities play important roles in sustainable development and are important in tourism organizations' strategies [2,13]. In addition, CRS not only can affect organizational operations but also can have positive impacts on the consumers' perceptions of the organizations themselves and on sustainability of the tourism industry.

In addition, collaboration among private firms, governmental organizations, and other non-governmental organizations is a key factor in developing CSR policies for the long-term success of sustainable tourism development because synergy and shared values among these parties are crucial for successful CSR implementation [5,7,13]. Moreover, CSR initiatives normally represent collaboration among various stakeholders, including private firms, public organizations, non-government organizations and local communities. Collaboration among these tourism-related organizations can shape their shared goals to ensure the expected outcomes, effective CSR planning and the efficient utilization of tourism resources.

Tourism organizations should always consider not only private benefits from tourism activities but also social benefits in the long-term development of sustainable tourism [6,14,15].

Some authors found that CSR strategies are crucial for companies as they offer great opportunities to contribute to society and allows companies to distinguish themselves from other competitors [16], especially when companies integrate the long-term implementation of CSR into their vision and mission [17].

Some authors also noted that the implementation of CSR constitutes an important activity for improving the level of awareness with respect to CSR activities, values, and strategies throughout a firm [16]. In this sense, CSR implementation not only includes taking action to achieve CSR outcomes but also includes communicating CSR activities to both stakeholders within the firms and those outside of the organizations [18]. It is obvious that successful CSR projects in tourism and also in other industries must pay attention to every step of corporate social responsibility from CSR planning and CSR activities, to CSR implementation and the measurement of CSR outcomes.

*1.1. Scope of Corporate Social Responsibility*

The scope of corporate social responsibility is presented in Figure 1. In terms of corporate sustainability, an organization demonstrates their corporate responsibility in three dimensions: social responsibility, economic responsibility, and environmental responsibility. In addition, all of an organizations' actions need to focus on their impacts on the planet, people, and profit (also known as the three pillars of sustainability or Triple Ps) [19,20]. Maintaining these three impacts in the implementation of a company's plans and in their vision, mission, and strategies is thus highly important.

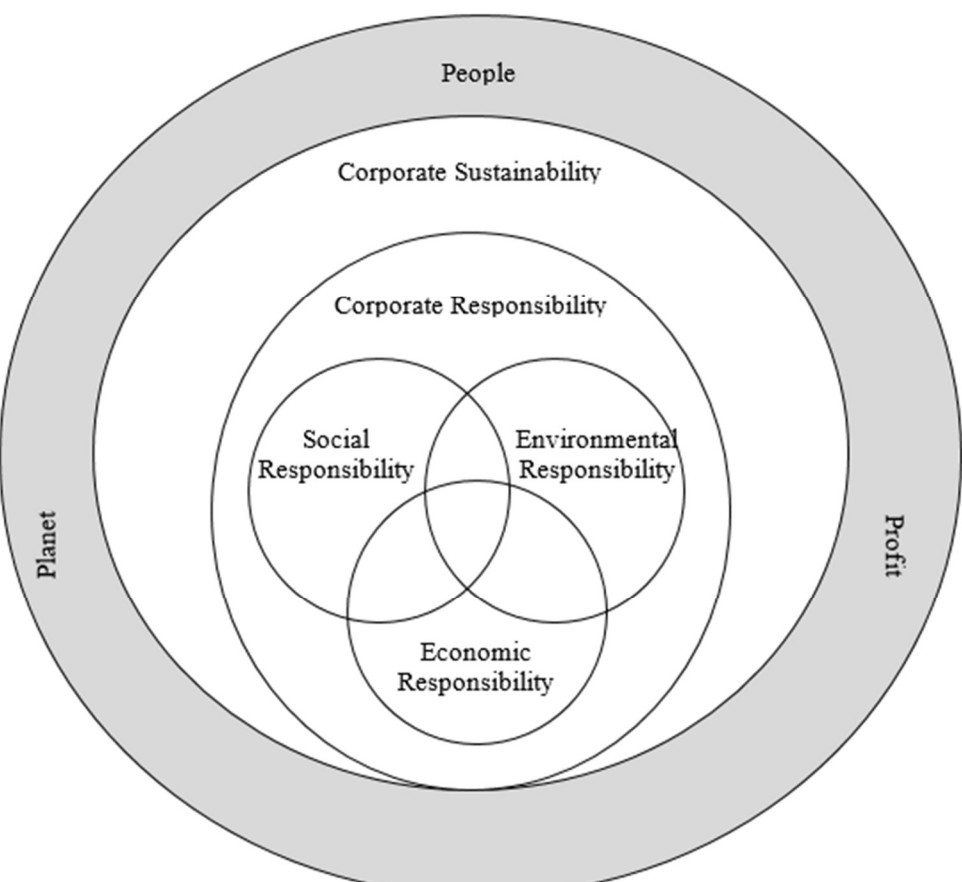

**Figure 1.** Scope of Corporate Social Responsibility. Source: Adapted from Marrewijk, V. M. (2003) [21].

The concept of corporate social responsibility has been developed through the lens of several theories, including the theory of the firm, resource-based view theory, institutional theory, agency theory, stakeholder theory, and stewardship theory [22].

Based on a review of articles published from 1992 to 2002, important topics related to the field of corporate social responsibility include environmental issues and ethics [23].

The benefits of corporate social responsibility include minimizing costs and corporate risks, improving legitimacy and corporate reputation, creating competitive advantages, and creating new values via the development of synergy within the organization [24].

One of the fast-growing aspects of corporate social responsibility in tourism is the identification of effective ways and methods of measuring corporate social responsibility performance, including reporting and external assurance for tourism and hospitality organizations [25]. The measurement of corporate social responsibility in tourism is one of the most important tools for developing this field of research, and measurements should be included at all levels, from the organizational level to the employee and consumer levels [26,27].

*1.2. Conceptualizing Corporate Social Responsibility in Tourism*

A large number of studies have been conducted and has set up a strong foundation for research in corporate social responsibility in tourism. In this review, data from research articles and journals dated between 2002 and 2022 are included. The following research questions represent the main focuses of this paper:

➢ Research question no. 1: What are the characteristics of scholarly work on corporate social responsibility in tourism published from 2002 to 2022?
➢ Research question no. 2: Which journals, authors, and articles on corporate social responsibility in tourism have achieved the greatest scholarly impact?
➢ Research question no. 3: What is the intellectual structure of research on corporate social responsibility in tourism?

## 2. Materials and Methods

This review adopted a science mapping approach based on bibliometric analyses, which is a process for documenting and synthesizing knowledge in the field of corporate social responsibility in tourism. Using this approach, this review offers an intellectual composition of knowledge developments in the field of corporate social responsibility in tourism. The evolution of this field and its other important dimensions are presented in maps or tables. The results of this methodology provide a clearer understanding of the topic, as well as identifies the key authors, influential articles, and the leading academic journals in the field. Therefore, the findings help guide new directions in research.

*Search Criteria and Identification of Sources*

The data used in this review were obtained from the Scopus database. Scopus is considered one of the most respected and widely used databases in social science research studies [28,29]. The time frame ranged from 2002 to 2022 (August), providing approximately 20 years of research in the field of corporate social responsibility in tourism. As the tourism industry is an important sector for many countries, including developed and developing countries, the focus of corporate social responsibility in tourism has expanded across many different fields of study, including tourism, hospitality, economic development, the environment, society, and cultures.

To develop an effective review process, the PRISMA guidelines (preferred reporting items for systematic reviews and meta-analyses) were used to search for and to identify articles in the Scopus database [30], as shown in Figure 2. The authors used (TITLE-ABS-KEY ("corporate social responsibility in tourism") AND TITLE-ABS-KEY (CSR in tourism)) in the keyword search, resulting in 1332 articles that were appropriate for inclusion in the data analysis. To implement the bibliometric analysis effectively, the authors

followed the protocol closely to ensure proper data analysis planning and execution, and transparency [31,32].

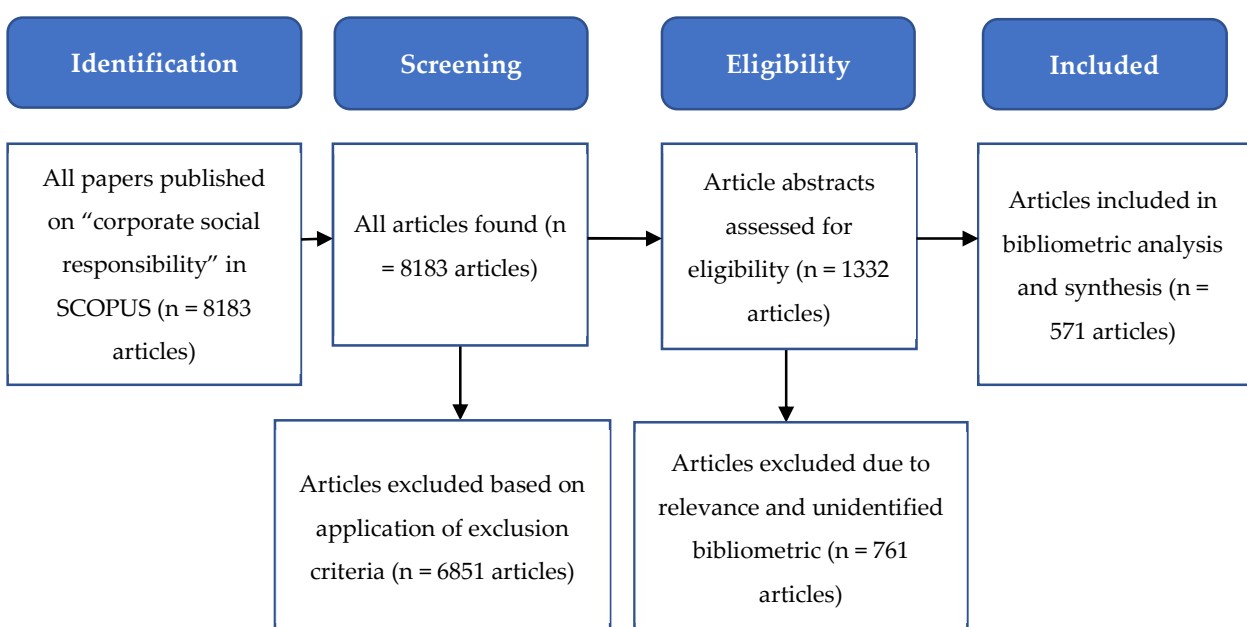

**Figure 2.** The preferred reporting items for systematic reviews and meta-analyses (PRISMA) flow diagram detailing the steps in identifying and screening sources.

Most of the search results were from journal articles, conferences, and books. Moreover, the search results included sources with data from the relevant fields and mainly contained academic work published in English. The last step in the data preparation was screening and verifying the articles' eligibility, especially for duplications and article types. Finally, 571 articles qualified for the bibliometric analysis.

## 3. Results

### 3.1. Data Extraction and Analysis

Descriptive statistics methods were mainly employed to analyze the geographic distribution, size, trajectories, and growth of data in the articles included in the study. In addition, for the main types of analyses, the authors applied several techniques, including citations and co-citations [33–35]. The authors used descriptive statistics to investigate the size, growth trajectory, and geographic distribution of articles in the review database [36,37] and applied citation and co-citation analyses to evaluate the roles and impacts of article titles and author names in the area of corporate social responsibility in tourism.

For the citation analysis, the influence of the articles in the selected database are studied by focusing on how frequently each article was cited by other authors or articles [38,39]. In addition, citation analysis showed the quality of the academic work from the viewpoints of other researchers.

Additionally, co-citation analyses involve evaluating how many times two articles or two authors are included in the references of the articles included in the current review [40–42]. Co-citation analysis is also known as an approach to help indicate the importance of a particular study in terms of interdisciplinary ideas.

Authors or articles being co-cited by other researchers represents intellectual similarity, showing the significant impacts of those authors or articles [43,44]. Another type of analysis used in the current study was keyword co-occurrence, which involves exploring the number of times a keyword co-occurs in the Scopus database. This type of analysis can highlight the changes in the focus of research in the field of corporate social responsibility in tourism over time, and the trends can provide future research directions in the field [36].

### 3.2. Review Findings

Table 1, shown below, highlights that research on corporate social responsibility in tourism is truly interdisciplinary by nature because the topic has been discussed in various journals and has focused on different themes, namely business management, tourism, hospitality, sustainability, the environment, and energy. In the analysis, the threshold for minimum number of articles required was 5 out of 248 sources, and 24 journals met the threshold. Tourism Management had the most citations for corporate social responsibility in tourism, with 1975 citations, followed by the International Journal of Contemporary Hospitality Management, with 1078 citations. In addition, the total link strength indicates the total strength of the citation links between a given journal and other journals.

**Table 1.** Ten major journals focusing on corporate social responsibility in tourism.

| No. | Journal | Subjects | Articles | Citations | Total Link Strength |
|---|---|---|---|---|---|
| 1 | Tourism Management | Tourism and management | 22 | 1975 | 162 |
| 2 | International Journal of Contemporary Hospitality Management | Hospitality and business | 13 | 1078 | 89 |
| 3 | Journal of Sustainable Tourism | Tourism and sustainability | 30 | 831 | 109 |
| 4 | International Journal of Hospitality Management | Hospitality and management | 11 | 608 | 66 |
| 5 | Current Issues in Tourism | Tourism and business management | 16 | 445 | 45 |
| 6 | Journal of Travel Research | Travel and tourism | 8 | 337 | 34 |
| 7 | Journal of Cleaner Production | Strategy and business management | 6 | 304 | 16 |
| 8 | Sustainability | Environment and energy | 33 | 259 | 82 |
| 9 | Asia Pacific Journal of Tourism Research | Tourism and business management | 8 | 148 | 32 |
| 10 | Corporate Social Responsibility and Environmental Management | Corporate social responsibility and environment | 6 | 90 | 11 |

Table 2 presents the co-citations of sources in the selected research articles. These results support those of the citation analysis by highlighting the influence of each source or journal in this field of research. The results showed that Tourism Management (1159 co-citations) had the top co-citations, followed by the Journal of Business Ethics (1102 co-citations), the International Journal of Hospitality Management (1089 co-citations), and the Journal of Sustainable Tourism (908 co-citations). Interestingly, the Journal of Business Ethics was not within the top ten highly cited sources but ranked second in terms of most co-cited source. Furthermore, the total link strength indicates the total strength of the co-citation links between a given journal and other journals.

**Table 2.** Top ten co-citations of sources in articles on corporate social responsibility in tourism, 2002–2022.

| Rank | Sources | Co-Citations | Total Link Strength |
|---|---|---|---|
| 1 | Tourism Management | 1159 | 42,373 |
| 2 | Journal of Business Ethics | 1102 | 39,102 |
| 3 | International Journal of Hospitality Management | 1089 | 46,969 |
| 4 | Journal of Sustainable Tourism | 908 | 32,519 |
| 5 | Annals of Tourism Research | 626 | 22,032 |
| 6 | International Journal of Contemporary Hospitality Management | 608 | 27,407 |
| 7 | Journal of Cleaner Production | 442 | 14,544 |
| 8 | Academy of Management Review | 338 | 13,188 |
| 9 | Journal of Business Research | 293 | 13,495 |
| 10 | Corporate Social Responsibility and Environmental Management | 277 | 10,875 |

In Table 3, the ten most highly cited articles on corporate social responsibility in tourism are presented. The top cited article had 568 citations [45], followed by the articles by Inoue and Lee, with 389 citations; by Sparks, Perkins, and Buckley [46,47], with 309 citations; and by Henderson, with 262 citations [48]. The authors also provided further analysis based on the research perspectives of the leading articles in the field of corporate social responsibility in tourism.

**Table 3.** Ten most highly cited articles on corporate social responsibility in tourism based on Scopus citations, 2002–2022 (*n* = 571), in order.

| Rank | Articles | Research Perspective | Citations | Links |
|---|---|---|---|---|
| 1 | Manaktola, K., & Jauhari, V. (2007). Exploring consumer attitude and behavior towards green practices in the lodging industry in India. International Journal of Contemporary Hospitality Management, 19(5), 364–377. | consumer perception | 568 | 9 |
| 2 | Inoue, Y., & Lee, S. (2011). Effects of different dimensions of corporate social responsibility on corporate financial performance in tourism-related industries. Tourism management, 32(4), 790–804. | CSR performance | 389 | 26 |
| 3 | Sparks, B. A., Perkins, H. E., & Buckley, R. (2013). Online travel reviews as persuasive communication: The effects of content type, source, and certification logos on consumer behavior. Tourism Management, 39, 1–9. | consumer perception | 309 | 1 |
| 4 | Henderson, J. C. (2007). Corporate social responsibility and tourism: Hotel companies in Phuket, Thailand, after the Indian Ocean tsunami. International Journal of Hospitality Management, 26(1), 228–239. | CSR review and policy | 262 | 26 |
| 5 | Font, X., Walmsley, A., Cogotti, S., McCombes, L., & Häusler, N. (2012). Corporate social responsibility: The disclosure–performance gap. Tourism Management, 33(6), 1544–1553. | CSR performance | 190 | 18 |
| 6 | Coles, T., Fenclova, E., & Dinan, C. (2013). Tourism and corporate social responsibility: A critical review and research agenda. Tourism Management Perspectives, 6, 122–141. | CSR review | 189 | 30 |
| 7 | Chou, C. J. (2014). Hotels' environmental policies and employee personal environmental beliefs: Interactions and outcomes. Tourism management, 40, 436–446. | Employee perspective | 179 | 5 |
| 8 | Zhu, Y., Sun, L. Y., & Leung, A. S. (2014). Corporate social responsibility, firm reputation, and firm performance: The role of ethical leadership. Asia Pacific Journal of Management, 31(4), 925–947. | CSR performance | 166 | 2 |
| 9 | Frey, N., & George, R. (2010). Responsible tourism management: The missing link between business owners' attitudes and behaviour in the Cape Town tourism industry. Tourism management, 31(5), 621–628. | Customer perspective | 159 | 8 |
| 10 | Theodoulidis, B., Diaz, D., Crotto, F., & Rancati, E. (2017). Exploring corporate social responsibility and financial performance through stakeholder theory in the tourism industries. Tourism Management, 62, 173–188. | CSR performance | 130 | 11 |

Figure 3 presents the research perspectives obtained based on the relationship among important keywords found in the top cited papers on corporate social responsibility listed in Table 3. These results highlight the focus of research in terms of evaluating corporate social responsibility from the perspectives of the consumer, employee, policies, and organization. In addition, the multidimensional perspectives on this topic can provide an improved understanding on the initiatives, development, and the implementation of corporate social responsibility in tourism.

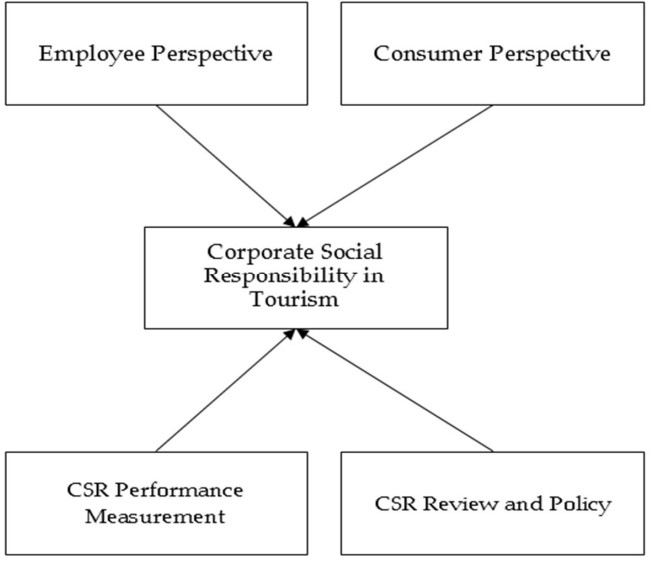

**Figure 3.** Research perspectives on CSR in tourism.

Table 4 below shows the co-citation analysis, presenting the degree or strength of the link for the articles that were co-cited within the 571 articles in this study. Remember, however, that these co-cited articles may be found in either the Scopus database or another academic database. In additional, these co-cited papers are considered by the authors or articles included in this review as very influential papers, implying that the journals of these highly co-cited papers are important in the field of corporate social responsibility in tourism: Tourism Management Review, International journal of hospitality management, Tourism Management, and Business Horizons.

**Table 4.** Ranking for the top ten most co-cited articles on corporate social responsibility in tourism, 2002–2022 (*n* = 571).

| Rank | Cited Reference | Citations | Total Link Strength |
|---|---|---|---|
| 1 | Coles, T., Fenclova, E., & Dinan, C. (2013). Tourism and corporate social responsibility: A critical review and research agenda. Tourism Management Perspectives, 6, 122–141. | 51 | 84 |
| 2 | Kang, K. H., Lee, S., & Huh, C. (2010). Impacts of positive and negative corporate social responsibility activities on company performance in the hospitality industry. International journal of hospitality management, 29(1), 72–82. | 30 | 75 |
| 3 | Inoue, Y., & Lee, S. (2011). Effects of different dimensions of corporate social responsibility on corporate financial performance in tourism-related industries. Tourism management, 32(4), 790–804. | 30 | 73 |
| 4 | Henderson, J. C. (2007). Corporate social responsibility and tourism: Hotel companies in Phuket, Thailand, after the Indian Ocean tsunami. International Journal of Hospitality Management, 26(1), 228–239. | 28 | 66 |
| 5 | Carroll, A. B. (1991). The pyramid of corporate social responsibility: Toward the moral management of organizational stakeholders. Business horizons, 34(4), 39–48. | 28 | 64 |
| 6 | Holcomb, J. L., Upchurch, R. S., & Okumus, F. (2007). Corporate social responsibility: what are top hotel companies reporting?. International Journal of Contemporary Hospitality Management, 19(6), 461–475. | 28 | 63 |
| 7 | Carroll, A. B., & Shabana, K. M. (2010). The business case for corporate social responsibility: A review of concepts, research and practice. International journal of management reviews, 12(1), 85–105. | 27 | 64 |
| 8 | Dahlsrud, A. (2008). How corporate social responsibility is defined: an analysis of 37 definitions. Corporate social responsibility and environmental management, 15(1), 1–13. | 24 | 47 |
| 9 | Garay, L., & Font, X. (2012). Doing good to do well? Corporate social responsibility reasons, practices and impacts in small and medium accommodation enterprises. International Journal of Hospitality Management, 31(2), 329–337. | 22 | 63 |
| 10 | De Grosbois, D. (2012). Corporate social responsibility reporting by the global hotel industry: Commitment, initiatives and performance. International Journal of Hospitality Management, 31(3), 896–905. | 22 | 60 |

In terms of the top countries (Table 5) publishing articles on corporate social responsibility in tourism, several countries from different continents showed a high number of citations in this field, including the United Kingdom (2305 citations) and Spain (1239 citations) in Europe and the United States (1622 citations) in North America.

**Table 5.** Top countries publishing research on corporate social responsibility in tourism from 2002 to 2022.

| Rank | Country | Citations | Articles | Total Link Strength |
|---|---|---|---|---|
| 1 | United Kingdom | 2305 | 69 | 618 |
| 2 | United States | 1622 | 72 | 466 |
| 3 | Spain | 1239 | 64 | 369 |
| 4 | Australia | 1095 | 46 | 178 |
| 5 | China | 784 | 52 | 316 |
| 6 | India | 688 | 14 | 93 |
| 7 | Canada | 489 | 13 | 173 |
| 8 | South Africa | 483 | 18 | 139 |
| 9 | Taiwan | 468 | 14 | 124 |
| 10 | Hong Kong | 348 | 19 | 165 |

One study highlighted that the majority of CSR research is mainly conducted in developed countries, including research reports on the environmental impacts and social benefits from CSR initiatives. However, from Figure 4, significant improvements in the amount of CSR research conducting in developing countries, including China, Malaysia, Turkey, Egypt, Saudi Arabia, India, and Pakistan, can be seen. In addition, Figure 3 demonstrates five clusters among the top countries with the most scholarly work on CSR in tourism. One of the most important highlights in Figure 4 is that each cluster includes countries from different regions or continents, implying that the scopes and issues of research on CSR in tourism has become more universal across continents. However, many existing studies on CSR in tourism primarily focus on advanced economies, and whilst environmental reporting is well established, research on the social impact of CSR and social reporting is either limited or absent [49].

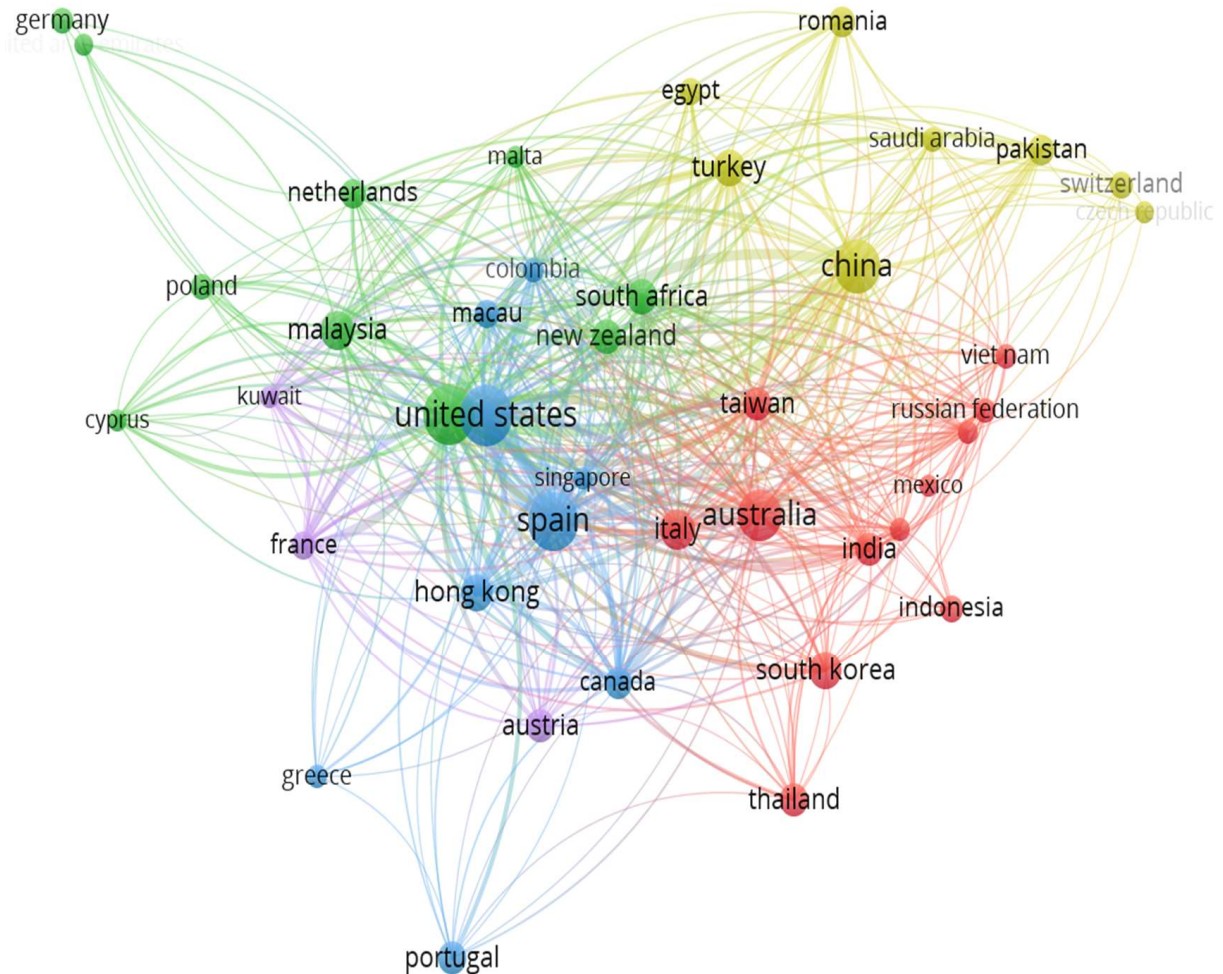

**Figure 4.** Map of the top countries focusing on corporate social responsibility in tourism.

The co-occurrence analysis presented in Figure 5 offers the structure of keywords with respect to corporate social responsibility in tourism and offers the relative focus of research based on the frequency of keywords. In addition, the keywords shown in the map are commonly used by researchers.

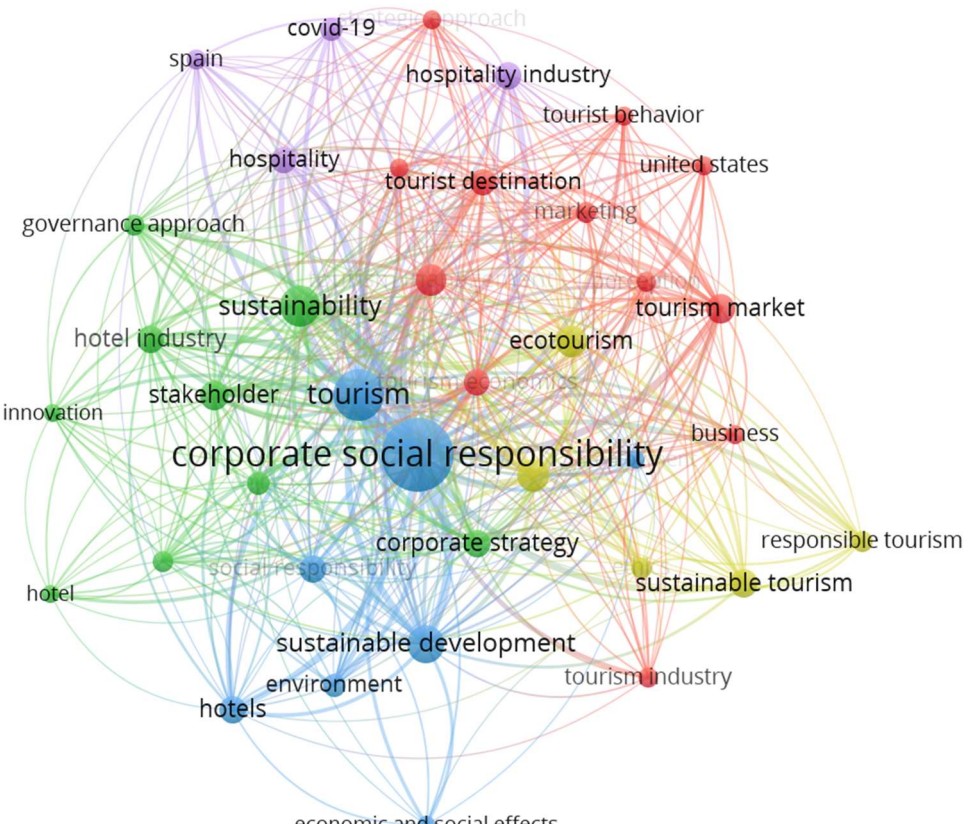

**Figure 5.** Co-word map for articles on corporate social responsibility in tourism published from 2002 to 2022 (threshold, 10 co-occurrences; displayed, 38 keywords).

The keywords in the study of CSR in tourism can be classified into five clusters. Cluster 1 includes keywords such as business, marketing, perception, strategic approach, and tourism economics. This shows that the research focus of CSR in tourism is in the areas of business management and strategic management.

Cluster 2 focuses on keywords such as sustainability, stakeholder, governance approach, and environmental management, representing another important cluster: research on policy related to CSR in tourism. Additionally, cluster 3 includes corporate social responsibility, economic and social effects, and sustainable development. Furthermore, cluster 4 contained ecotourism, ethics in tourism, and responsible tourism, showing that research on CSR focused on each type of tourism activity and practice. Finally, cluster 5 showed the aspects of CSR in the hospitality industry and in terms of the COVID-19 pandemic. This last cluster suggested a close relationship between tourism and hospitality and implied that even in a crisis, such as the COVID-19 pandemic, CSR issues are always an important concept to consider.

In terms of emerging themes in research on CSR in tourism (Figure 6), a greater number of scholars shifted their research in the early 2000s to 2006 from responsible tourism, tourism market, and corporate strategies to new research areas such as tourist behaviors; hotel industries; strategic approaches; and the COVID-19 pandemic, from 2018 to 2022. From 2020 to 2022, the COVID-19 pandemic clearly affected various areas of the tourism industry, including the scope of corporate social responsibility. Moreover, the emerging trends in research on corporate social responsibility in tourism have become a prominent strategic approach, implying that corporate social responsibility practices have been implemented in long-term plans and integrated into the visions and missions of tourism organizations.

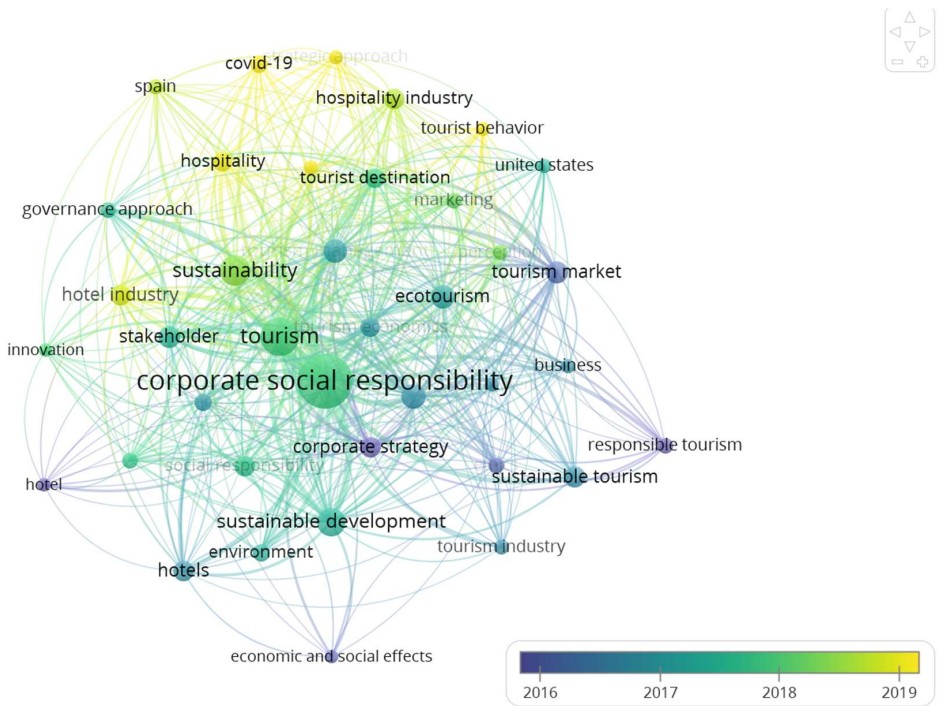

**Figure 6.** Co-word mapping of corporate social responsibility in tourism based on 571 articles from 2002 to 2022.

The results in Table 6 show that Font, X. led the number of citations, with 809 citations in topics related to CSR in tourism, followed by Lee, with 637 citations. Su contributed 5 articles with 219 citations; Su is the third most-cited author in this field. In addition, the table below presents the authors' various research areas as well, which include sustainability and tourism, corporate social responsibility, corporate philanthropy, and corporate governance and strategy.

**Table 6.** Top cited authors on corporate social responsibility in tourism (*n* = 571).

| Rank | Author | Affiliations | Research Area | Citation | Articles | Total Link Strength |
|------|--------|--------------|---------------|----------|----------|---------------------|
| 1 | Font, X. | University of Surrey, UK | Sustainability, marketing in tourism and hospitality | 809 | 9 | 32 |
| 2 | Lee, S. | Pennsylvania State University, United States | Corporate social responsibility | 637 | 10 | 26 |
| 3 | Su, L. | Central South University, China | Corporate social responsibility; corporate philanthropy | 219 | 5 | 7 |
| 4 | Uyar, A. | Excelia Business School, France | Corporate governance, sustainability reporting, corporate reporting | 88 | 6 | 31 |
| 5 | Karaman, A. S. | American University of the Middle East, Kuwait | Corporate social responsibility, sustainability | 85 | 5 | 27 |
| 6 | Kuzey, C. | Murray State University Murray disabled, United States | Corporate social responsibility performance | 85 | 5 | 27 |
| 7 | Camilleri, M.A. | University of Malta, Malta | Strategy, sustainable development, technology adoption | 43 | 5 | 11 |
| 8 | Ahn, J. | Hanyang University, South Korea | Brand personality; community participation | 37 | 5 | 3 |
| 9 | Manente, M. | Ca' Foscari University, Italy | Responsible tourism and corporate social responsibility (CSR) | 23 | 7 | 0 |
| 10 | Minghetti, V. | Ca' Foscari University, Italy | Responsible tourism and corporate social responsibility (CSR) | 23 | 7 | 0 |

The co-citation analysis based on cited authors used the minimum number of citations, with 40 and 118 authors meeting the threshold.

The co-citation analysis of authors in Table 7 highlights the important role of the authors with a high number of citations in confirming the impact and influence of corporate social responsibility in tourism. The results indicated that Lee, S. (437 co-citations) led in terms of ranking for co-citations in this field, followed by Font, X. (347 co-citations), Carroll, A.B. (257 co-citations), and Bohdanowicz, P. (156 co-citations). Obviously, these leading authors are also the top authors in the citation analysis, re-stating the importance of their work and contributions to the literature with respect to corporate social responsibility in tourism.

**Table 7.** High-impact scholars in the area of corporate social responsibility in tourism based on co-citations.

| Rank | Author | Co-Citations | Total Link Strength | Focus |
| --- | --- | --- | --- | --- |
| 1 | Lee S. | 437 | 14,856 | Corporate social responsibility |
| 2 | Font X. | 347 | 11,347 | Sustainable marketing in tourism |
| 3 | Carroll A.B. | 257 | 7136 | CSR; business ethics |
| 4 | Bohdanowicz P. | 156 | 5172 | Environment and sustainability |
| 5 | Bhattacharya C.B. | 148 | 4694 | Corporate responsibility, business ethics and sustainability |
| 6 | Sen S. | 148 | 4421 | Consumer behavior and sustainability |
| 7 | Okumus F. | 136 | 4371 | Tourism and hospitality |
| 8 | Han H. | 134 | 3624 | Sustainable tourism and hospitality marketing |
| 9 | Hall C.M. | 131 | 3311 | Tourism and sustainability |
| 10 | Kang K.H. | 120 | 4077 | Corporate governance and corporate social responsibility |

### 3.3. Intellectual Structure of Research on Corporate Social Responsibility in Tourism

Using the results of the co-citation analysis with respect to the authors, VOSviewer showed 30,223 authors from 571 research papers (Figure 7). The map generated by VOSviewer presented 118 author co-citations with a threshold of 40 co-citations.

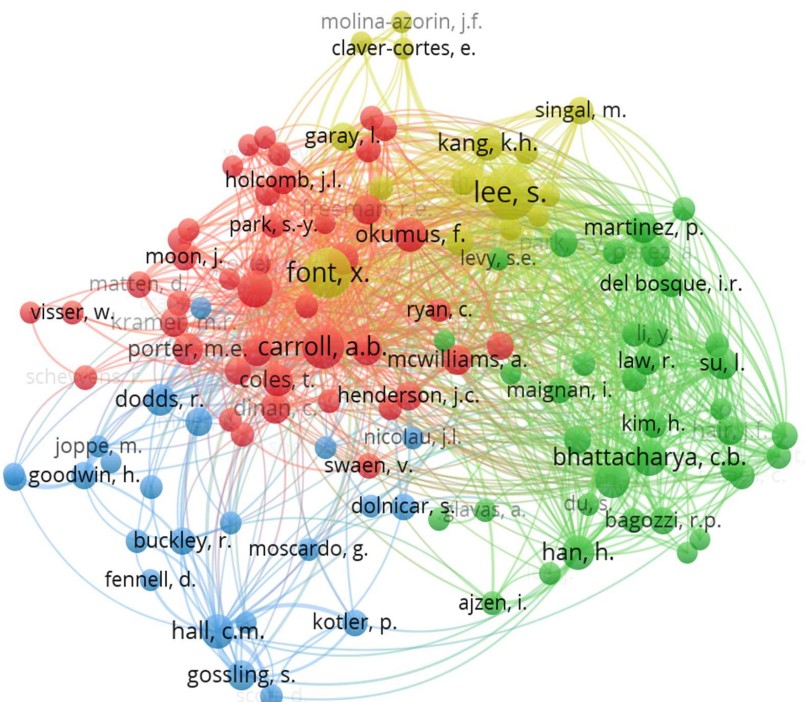

**Figure 7.** Co-citation analysis map for corporate social responsibility in tourism, 2002–2022 (threshold, 40; displayed, 118 authors).

The four schools of thought from the data are as follows: To start, the first school of thought can be considered corporate social responsibility performance, with the leading scholars being Lee (437 citations), Font (347 citations), and Kang (120 citations) (CSR performance (shown in yellow circles)). Next, corporate social responsibility policy is the

second school of thought, including scholars such as Carroll (257 citations), Bohdanowicz (156 citations), and Okumus (136 citations) (shown in red circles). The third school of thought includes business ethics and includes leading authors such as Bhatacharya (148 citations), Sen (148 citations), and Han (134 citations) (shown in green circles). Lastly, the theme of sustainable tourism and environment is represented in the literature on corporate social responsibility in tourism by important authors such as Hall (131 citations), Dodds (104 citations), and Gossling (91 citations) (sustainable tourism and environment (shown in blue circles)).

## 4. Discussion

The current review achieved the objectives mentioned in the Introduction; this paper presented new insights related to academic investigations into corporate social responsibility in tourism. Influential authors, articles, and journals were presented and discussed.

The findings of the current study directly respond to all of the research questions. For the first research question, the author offered scholarly work on corporate social responsibility in tourism published from 2002 to 2022, and the current review showed that corporate social responsibility in tourism has been explored using different points of view, including the perspectives of the consumer, employee, organization, and stakeholders. Research in these areas expanded mainly from countries in Europe, especially the United Kingdom, to several developing countries in the Asia Pacific regions and African countries.

Regarding research question no. 2, high scholarly impact journals in the context of corporate social responsibility in tourism included Tourism Management, International Journal of Hospitality Management, and Journal of Sustainable Tourism. In addition, influential authors and articles were discovered, presented in Tables 4–7.

For research question no. 3, the intellectual structure of research on corporate social responsibility in tourism was observed in four school of thoughts: corporate social responsibility performance, corporate social responsibility policy, business ethics, and sustainable tourism and environment. These four school of thoughts provide guidelines for and the focus of future academic studies in order to investigate each area in more depth and to explore the connection and relationship among these schools of thoughts.

The current study presented an important trend of increasing CSR research in developing countries by showing significant improvements in the number of CSR research studies in developing countries, including China, Malaysia, Turkey, Egypt, Saudi Arabia, India, and Pakistan. This new trend of academic work indicates that developing countries should study, monitor, investigate, and offer sustainable guidelines for CSR in tourism, which would help raise awareness and discussion among stakeholders in the tourism industry in order to support the appropriate CSR initiatives and to help avoid the negative impact of tourism business development in the long run.

In existing studies, based on a review of areas related to corporate social responsibility from 1992 to 2002, environmental issues and ethics were found to be important topics [23,50]. However, the current review showed that emerging research areas in corporate social responsibility, especially in the tourism areas, include strategic approaches to corporate social responsibility, the understanding of tourist behaviors, and hospitality management. The findings of this review introduced new dimensions and new focuses with respect to research in this field.

This review aimed to provide maps from past articles on the topic of corporate social responsibility in tourism. With a total number of 571 articles from the Scopus database, several bibliometric data analyses offered new highlights and findings that are useful for obtaining a deeper understanding and that are helpful for new researchers.

## 5. Conclusions

### 5.1. Interpretation of the Findings

This review was conducted based on strict data analysis guidelines, and a total of 571 articles in the field of corporate social responsibility in tourism were thoroughly

selected and analyzed in this study. In addition, the number of studies on corporate social responsibility in tourism has increasing become significant in the past two decades, indicating the significance of this field as a topic of interest in the research community.

As shown in the results, the geographical distribution of research studies in the field of corporate social responsibility in tourism expanded beyond developed countries toward developing countries, including China, showing that the gap in the literature has been narrowed, thus meaning that the acceptance of or interest in corporate social responsibility with respect to tourism organizations has become a norm in corporate practices. In addition, several academic journals and conferences have increasingly addressed or proposed themes or Special Issues related to the field of corporate social responsibility in tourism.

Furthermore, the leading journals publishing studies on corporate social responsibility in tourism included the top journals in tourism research: such as Tourism Management, International Journal of Contemporary Hospitality Management, Journal of Sustainable Tourism, and International Journal of Hospitality Management. Interestingly, this review discovered that the topic of corporate social responsibility in tourism has also been highlighted in regional journals (e.g., Asia Pacific Journal of Tourism Research and African Journal of Hospitality, Tourism and Leisure), demonstrating growing interest with respect to this topic in various parts of the world.

Additionally, this review showed that the evolution of research on corporate social responsibility in tourism started in responsible tourism but has moved towards performance or outcomes of corporate social responsibility, and the new focus will be on understanding changing tourist or consumer behaviors as part of tourism organizations' corporate social responsibility.

### 5.2. Limitations of the Current Study

This review is not without limitations. First, the data used in this study were obtained only from the Scopus database; though this database is known as one of the largest academic databases in the world and many articles are included in Scopus, some articles on corporate social responsibility in tourism are only available in other databases.

Second, the co-citation analysis, which is one of the main techniques used in this review, cannot be directly interpreted and requires substantial knowledge of the subject for a better understanding of the results of this type of analysis [51].

### 5.3. Directions for Future Research

Even though this study used only Scopus, which is one of the database with the widest coverage of academic work, future research should include the Web of Science (WOS) or other databases, which may present additional dimensions and some other highlights in the field of corporate social responsibility in tourism. Another suggestion for future research work is a comparative investigation of the concept of corporate social responsibility in tourism in specific regions, such as Asia and Europe, which could help researchers to discover more in-depth and region-specific dimensions; therefore, the recommendations and implications for policy implications would be more applicable to the issues found in those regions. Moreover, future research may focus more on the impact of the COVID-19 pandemic on CSR in tourism as the results of this review showed that understanding the dimensions of crisis management (e.g., COVID-19) and providing a more in-depth analysis should lead to a greater understanding of corporate social responsibility.

**Author Contributions:** C.Y. conceptualized the research project; C.Y., S.N. and B.K. participated in the literature review; C.Y. and S.N. collected and analyzed the data; C.Y., S.N. and B.K. prepared and finalized the manuscript. All authors have read and agreed to the published version of the manuscript.

**Funding:** This research was supported by the ASEAN Centre for Sustainable Development Studies and Dialogue (ACSDSD) and the College of Management, Mahidol University.

**Institutional Review Board Statement:** This study was conducted according to the guidelines of the Declaration of Helsinki and was approved by the Central Institutional Review Board of Mahidol University.

**Informed Consent Statement:** Informed consent was obtained from all subjects involved in this study.

**Data Availability Statement:** The data presented in this study are available on request from the Corresponding author. The data are not publicly available due to the privacy of informants as several of them are government officers.

**Acknowledgments:** The authors thank the editors and all reviewers for their valuable comments and advice for improving this paper and thank the research participants for their participation in this study.

**Conflicts of Interest:** The authors declare no conflict of interest.

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
