# Peer review of "Bibliometric Analysis of Corporate Social Responsibility in Tourism"

_sustainability, doi:10.3390/su15010668_

Round 1

Reviewer 1 Report

I appreciate the authors’ efforts in the work concerning bibliometric analysis of corporate social responsibility in tourism. The paper is well structured and it provides an interesting analysis of the topic:

-        The information is relatively easy to navigate, and the structure of the paper allows readers to analyze the concepts approached.

-        The literature review provides a good background of the domain.

-        The authors bring relevant and interesting arguments to the investigated field.

However, authors could improve the article by paying attention to the following issues:

(1)   The information should be better structured, as some paragraphs are formed of single phrases. It is advised that paragraphs include more than one sentence/phrase related to the same idea.

(2)   Authors use bibliometric analysis to investigate corporate social responsibility in tourism, but they use PRISMA framework, which is used for systematic reviews. Instead of PRISMA framework, authors are advised to develop a research methodology framework applied for bibliometric review. In this direction, authors are advised to study and cite some updated works such as the following:

-       Exploring the Research Regarding Frugal Innovation and Business Sustainability through Bibliometric Analysis. Sustainability. 2022, 14(3), 1326. https://doi.org/10.3390/su14031326.

-        Bibliometric Analysis of the Green Deal Policies in the Food Chain. Amfiteatru Econ. 2022, 24, 410–428. DOI:10.24818/EA/2022/60/410.

In this manner, the reference list could be enhanced, as now it is limited.

(3)   The tables in the article need to be formatted according to the requirements of the journal.

(4)   The research presented in the study has been carried out using Scopus bibliographic database. Why has the Web of Science database, regarded as more comprehensive by some scientists, not been used?

(5)   Authors are strongly recommended to do some editing of their revised draft (for example: “This type of analysis can highlight changing research focus trends in in the field of corporate social responsibility in tourism” – line 191).

(6)   The reference list should be formatted according to the requirements of the journal.

(7)   Authors also need to include information regarding future research directions. Although, they mention about it in the abstract, there is no distinct section or paragraph in the article that covers future research directions.

Good luck with the revision!

Author Response

The authors were revised to improve, as suggested by reviewer.

Reviewer 2 Report

I suggest adding the complete list of publications analyzed as an annex.

Be aware that there are Thai fonts in Table 1.

Table formatting does not follow the journal's standard.

It would be helpful to explain the meaning of "total link strength." This expression appears in all tables but is not explained in the text.

Figures 3 and 4 are graphically attractive but are too little described. We don't know the meaning of the different colors used (if any). And what do the links (lines) represent? Regarding the keywords, I am guessing the links connect those keywords that appeared together in the research. But what are the links between different countries? Figure 5 is already clearer, thanks to the colored timeline in the legend. Figure 6 is also more understandable because the text explains the colors - but do the links represent citations? I suggest capitalizing the names of the authors.

The first paragraph of the discussion is a copy-paste of the journal's template. Do not forget to erase it.

The discussion should be enriched with a view on future research perspectives regarding corporate social responsibility and referenced in the secondary literature.  The interest in reviewing existing knowledge is to show missing aspects and designate new research perspectives. Same with the conclusions: describing the current state of the knowledge should be followed by suggesting new directions. 

Author Response

(The authors gave the same response as above.)

Reviewer 3 Report

Dear authors:

The manuscript reviewed the extant literature on corporate social responsibility to the development process of related fields, current research hotspots, and future research directions. The literature review on corporate social responsibility in tourism is necessary and valuable. However, there are several issues that the authors need to consider.

The main issue of the manuscript is the lack of story-telling ability. This mainly can be reflected in the introduction section and analysis results section. For example, the introduction section's content and construct didn't explain this study's value and significance logically, which I think cannot grasp readers' eyes. In addition, the cited references do not seem to blend with the content of the authors' arguments well, resulting in less convincing analyzed results. Furthermore, I'm afraid the authors failed to extract further valuable and meaningful conclusions based on the literature review results.

Besides, it would be better if the authors improved the abstract, including the factors like study aim, study content, study method, and results and conclusions. It will also be more acceptable if the authors reconsider the keywords.

In the Search Criteria and Identification of Sources part, I recommend the authors show the selection criteria of the studies in detail.

At last, some format and expression problems need the authors' attention.

Author Response

(The authors gave the same response as above.)

Round 2

Reviewer 2 Report

The article has been essentially improved. I don't have any other substantial doubts or remarks. I suggest proofreading for some minor errors in English. 

Author Response

The authors was revised and to improve, as suggested by Reviewer.

Reviewer 3 Report

Dear authors:

We can see the authors have followed the advice carefully, and I appreciate the authors' efforts in revision. The manuscript has improved a lot, but at the same time, there remain issues that didn't well solve in the last revision.

Firstly, the statement in the introduction section is still not convincing. You can fix this problem by rephrasing the sentence or reconstructing the framework more logically and convincingly. Also, the cited references do not seem to blend well with the authors' arguments' content. And as a review, the number of cited studies in this manuscript is small. You can add some in the result section.

Secondly, it would be better if the authors polished the statements of the manuscript.

Author Response

The authors was revised to improve, as suggested by Reviewer.
